# Comparative proteomics of HepG2 cells reveals NGLY1 as an important regulator of ferroptosis resistance and iron uptake

Stuart Emmerson[1], Haruhiko Fujihira[1,2]*, Takehiro Suzuki[3], Naoshi Dohmae[3], Peter Greimel[4], Yoshio Hirabayashi[5,6], Tadashi Suzuki[1]*

1 Glycometabolic Biochemistry Laboratory, Pioneering Research Institute, RIKEN, Wako, Saitama, Japan, 2 Division of Glycobiologics, Juntendo University Graduate School of Medicine, Bunkyo-ku, Tokyo, Japan, 3 Biomolecular Characterization Unit, Technology Platform Division, RIKEN Center for Sustainable Resource Science, Wako, Saitama, Japan, 4 Laboratory for Cell Function Dynamics, RIKEN Center for Brain Science, Wako, Saitama, Japan, 5 Institute for Environmental and Gender-Specific Medicine, Juntendo University Graduate School of Medicine, Urayasu, Chiba, Japan, 6 Cellular Informatics Laboratory, Pioneering Research Institute, RIKEN, Wako, Saitama, Japan

* haruhiko.fujihira@riken.jp (HF); tsuzuki_gm@riken.jp (TS)

## Abstract

NGLY1 deficiency is a rare genetic disorder caused by mutations in the *NGLY1* gene. This disorder presents a wide range of clinical symptoms, and its severity varies among affected individuals. Previous studies have focused on understanding the influence of NGLY1 on energy metabolism, revealing dysregulation in lipid metabolism following *NGLY1* deletion. In this study, we investigated the consequences of the loss of *NGLY1* on ferroptosis and iron homeostasis using human hepatocellular carcinoma cells, HepG2. Comparative proteomics analysis revealed significant alterations in protein quantities in *NGLY1*-deficient HepG2 cells, indicating that these cells are under "pro-ferroptotic" stress state. Moreover, dysregulated iron uptake and increased reactive oxygen species production were observed in the absence of NGLY1, indicating a novel perspective on the consequences of the loss of *NGLY1*. These findings provide important insights into the molecular pathways affected by *NGLY1* deletion and may contribute to the development of potential therapeutic strategies for individuals with NGLY1 deficiency.

## Introduction

NGLY1 deficiency is a rare genetic disorder resulting from mutations in the *NGLY1* gene, leading to significant disruptions in cellular metabolism [1–5]. Peptide:*N*-glycanase (PNGase, NGLY1 in mammals) is a highly conserved enzyme with a primary role in cellular protein quality control through the endoplasmic reticulum (ER)-associated degradation (ERAD) pathway, which targets misfolded glycoproteins for degradation [6]. Although nuclear factor erythroid 2-related factor 1 (NRF1) is the

**Data availability statement:** The raw data of proteomics analyses are available from PRIDE (https://www.ebi.ac.uk/pride/, Project accession: PXD058551).

**Funding:** This work was supported by RIKEN Pioneering Project ("Glyco-Lipidologue Initiative") (to TadS), Japan Agency for Medical Research and Development-Core Research for Evolutional Science and Technology (AMED-CREST, JP24gm14100003) (to TadS and HF). The funders had no role in study design, data collection and analysis, decision to publish, or preparation of the manuscript.

**Competing interests:** The authors have declared that no competing interests exist.

most well-characterized substrate [7], NGLY1 likely targets a broad range of misfolded glycoproteins in the cytosol. By cleaving the amide bonds between *N*-glycans and the asparagine residues of proteins, which facilitates their degradation by proteasomes [2,4–6], this process is essential to prevent misfolded protein accumulation and potential cytosolic aggregation, thereby helping to maintain cellular homeostasis. The absence of functional NGLY1 in individuals with this autosomal recessive disorder results in a constellation of symptoms, including developmental delays, movement disorders, intellectual disabilities and impaired liver function, to name a few [6]. The severity of NGLY1 deficiency is different for each patient, with the wide breadth of personalized care required for treatment standing as testimony to the significance of NGLY1's role in human metabolism and the complexities of managing NGLY1 deficiency [4,8]. The inconsistent combination of symptoms exhibited by sufferers of this condition has made predictable and consistent treatment difficult, as there remains much to be understood about the full array of pathways affected by the absence of NGLY1, and so personalized care is often required to address the disorder's diverse symptoms.

Some of these questions were explored previously [9] in a more organ-specific manner by focusing on the effects of NGLY1 deletion on liver metabolism, a central hub for energy homeostasis, using a liver-specific *Ngly1*-knockout (KO) mouse model. On a normal rodent diet, liver-specific *Ngly1*-KO mice exhibited little difference compared to their control littermates. However, following a high fructose diet (used to model non-alcoholic fatty liver disease (NAFLD) [10]), significant dysregulation in the metabolism of lipids emerged in liver-specific *Ngly1*-KO mice [9]. While the precise mechanisms by which NGLY1 exerts its regulatory control over lipid metabolism in the liver remains poorly understood, these results demonstrated the dynamic role that NGLY1 plays in energy metabolism and the potentially wide network of its influence.

A link between NGLY1, ferroptosis upregulation and NRF1 has been reported previously [7], though the cellular mechanisms and NGLY1's full role remained vaguely understood. Ferroptosis is a regulated form of cell death characterized by the iron-dependent accumulation of lipid peroxides [7,11,12]. To investigate the consequences of loss of *NGLY1* on cellular homeostasis and lipid metabolism, we generated *NGLY1*-KO HepG2 (human hepatocellular carcinoma cells) cell lines. *NGLY1*-KO and wild type (WT) HepG2 cells revealed new insights into the altered landscape of the cellular proteome in the absence of NGLY1, particularly in the direction of cellular ferroptosis. *NGLY1*-KO HepG2 cells were found to be in a state of pro-ferroptosis stress. In addition, a marked dysregulation in iron uptake, likely owing to heightened presence of transferrin (TF) and transferrin receptor (TFRC/CD71) in *NGLY1*-deficient cells, was also observed. It seems plausible that this elevated iron influx resulted in the formation of significant quantities of reactive oxygen species (ROS), as evidenced in an *NGLY1*-deficient cellular environment. The findings presented here illuminate the possible regulatory influence of NGLY1 over iron metabolism and, by extension, the ROS environment within human cellular systems. Furthermore, the possibility of ferroptosis as a key cellular process in NGLY1 deficiency could open

new avenues for targeted interventions aimed at regulating iron homeostasis and ROS levels, potentially offering therapeutic approaches for individuals affected by this rare disorder.

## Materials and methods

### Antibodies

Antibodies against NGLY1 and GPX4 were purchased from Sigma Aldrich (#HPA036825) and abcam (#ab125066), respectively. Antibodies against GAPDH and β-actin were purchased from Merck Millipore (#MAB374) and Cell Signaling Technology (#3700), respectively. Horseradish peroxidase (HRP) conjugated secondary antibodies were purchased from Jackson ImmunoResearch (Peroxidase AffiniPure Donkey Anti-Rabbit IgG (H + L), #711-035-152; Peroxidase AffiniPure Donkey Anti-Mouse IgG (H + L), #715-035-151).

### Cell culture and treatment

WT and *NGLY1*-KO HepG2 cells were cultured in DMEM-low glucose (LG) (1.0 g/L D-(+)-Glucose) (Nacalai Tesque, #08456−65) supplemented with 10% fetal bovine serum (FBS) (Nichirei Biosciences, #175012). WT and *NGLY1*-KO/*Ngly1*-KO HEK293T or MEF cells were cultured in DMEM-high glucose (HG) (4.5 g/L D-(+)-Glucose) (Nacalai Tesque, #08457−55) supplemented with 10% FBS and 100 unit/mL penicillin and 100 μg/mL streptomycin. Cells were maintained in an incubator at 37°C with 5% $CO_2$. For general maintenance the culture medium was refreshed every 48 hours and cells were passaged at approximately 80% confluency. The isolation of MEF cells were previously described [13].

### Knockout of *NGLY1* in HepG2 and HEK293T cells

To generate *NGLY1*-KO HepG2 cells by CRISPR/Cas9-mediated gene editing, guide sequences targeting *NGLY1* were designed and cloned into pSpCas9(BB)-2A-Puro (Addgene, #62988) vector according to a previous report [14]. The following targeting guide sequence of *NGLY1* was used: 5'- GGCCTCCAAAAAGGTCTCCG-3'. HepG2 and HEK293T cells were transfected with the generated pSpCas9(bb)-2A-Puro-NGLY1, followed by puromycin (1.0 μg/ml) selection for several days. *NGLY1*-KO clones were obtained by limited dilution, and the *NGLY1*-KO was examined by immunoblotting. *NGLY1*-deletion was also confirmed by genome sequencing. In this study, two isolated clones of *NGLY1*-KO HepG2 cells, named KO-1 and KO-2, were utilized. Regarding *NGLY1*-KO HEK293T cells, one isolated clone was used.

### Comparative proteomics

To investigate the differences in protein expression between *NGLY1*-KO and WT HepG2 cells, comparative proteomics analysis was performed as previously reported [15]. In this experiment, we used one of the obtained *NGLY1*-KO clones, specifically KO-1. Three technical replicates of KO-1 and WT cell lines were trypsinized, counted and $2 \times 10^6$ cells were used for each replicate sample for analysis. Cell lysates were prepared using RIPA buffer (50 mM tris(hydroxymethyl) aminomethane (Tris)-HCl (pH 8.0), 150 mM sodium chloride, 1.0% Nonidet p-40 (NP-40), 0.5% sodium deoxycholate, 0.1% sodium dodecyl sulfate (SDS)), and the protein concentration was determined using a Bio-Rad Protein Assay kit (#500–0006). The protein samples were reduced with 1 M Tris-HCl (pH 8.0), 7 M guanidine-HCl, 10 mM EDTA, 50 mM DTT followed by carboxymethylation with iodoacetate. The protein solutions were desalted using PAGE Clean Up Kit (Nacalai Tesque, #06441–50) and digested with TPCK-treated trypsin (Worthington Biochemical Corporation). The peptide concentration was determined using amino acid analysis [15]. Equal amounts of peptides were separated with a reversed phase nano-spray column (NTCC-360/75-3-105, NIKKYO technos) using Easy nLC 1200 (Thermo Fisher Scientific) and then applied to Q Exactive HF-X hybrid quadrupole-Orbitrap mass spectrometer (Thermo Fisher Scientific). MS and MS/

MS data were obtained with TOP10 method. The label free quantification was carried out using Proteome Discoverer 2.4 (Thermo Fisher Scientific) and Mascot (version 2.8.0, Matrix Science). The MS/MS data were used to search the NCBI-nr protein database (Taxonomy:human), using the following parameters: enzyme = trypsin; maximum missed cleavages = 3; variable modifications = Acetyl (Protein N-term), Gln->pyro-Glu (N-term Q), Oxidation (M), carboxymethyl (C); peptide mass tolerance = ± 15 ppm; fragment mass tolerance = ± 30 mmu; instrument type = ESI-TRAP. The acquired data were analyzed using QIAGEN IPA Software to identify differentially expressed proteins and reveal biological pathways associated with the observed changes.

## Immunoblotting

Protein expression levels were evaluated by western blotting. Cell lysates were prepared using RIPA buffer, and the protein concentration was determined using BioRad Protein Assay Kit. Ten µg of protein were loaded onto SDS-PAGE gels, separated by electrophoresis, and transferred onto PVDF membranes. The membranes were blocked with Bullet Blocking One blocking reagent (Nacalai Tesque, #13779−01), washed three times with PBS containing 0.05% Tween 20 (PBST) and then incubated with primary antibodies targeting the proteins of interest at 1:1,000 dilution within PBST at room temperature for one hour or overnight at 4°C. Subsequently, membranes were washed three times with PBST and then probed with appropriate secondary antibodies conjugated with HRP at room temperature for 45 minutes, before three washes of PBST. Protein bands were visualized using an enhanced chemiluminescence system (Merck Millipore, #WBKLS0500) and Fusion Solo S imaging system (VILBER). Densitometric analysis of immunoblot bands was conducted using ImageJ software (version 1.54K, National Institutes of Health, Bethesda, MD, USA). To quantify target protein expression, the intensity of each target band was measured and normalized against the intensity of the corresponding β-actin band. Briefly, bands were selected using the rectangular selection tool, and integrated density values were obtained. Ratios of target protein to β-actin were calculated to assess relative protein expression, normalizing for loading variations. Paired t-tests were conducted to compare relative expression levels between WT and *NGLY1*-KO HepG2 samples.

## RNA collection and qPCR

Total RNA was collected from adhesive cells using a commercially available RNA extraction kit and as per the manufacturer's instructions (QIAGEN RNeasy® Mini Kit, #74104). The quality and quantity of RNA were determined using a spectrophotometer (Denovix). Complementary DNA (cDNA) synthesis was performed using Superscript IV First-Strand Synthesis System (Invitrogen, #18091050) with Oligo(dT) 20. Quantitative PCR (qPCR) was conducted to measure the mRNA expression levels of GPX4 using the forward primer (ACAAGAACGGCTGCGTGGTGAA) and reverse primer (GCCACACACTTGTGGAGCTAGA), and a SYBR-green-based detection system (PowerUP SYBR Green Master Mix, Applied Biosystems, #A25742) using a QuantStudio5 (Applied Biosystems). The relative expression levels were calculated using GAPDH as housekeeping gene with forward primer (GTCTCCTCTGACTTCAACAGCG) and reverse primer (ACCACCCTGTTGCTGTAGCCAA).

## Cellular ROS assay

To assess the cellular ROS levels, the DCFDA/H2DCFDA Cellular ROS Assay Kit (abcam, #ab113851) was employed as per the manufacturer's instructions, with ensuing fluorescent intensity measured at Ex/Em = 485/535 nm using a Synergy H1 microplate reader (BioTek). Fluorescent microscopy using Cellular ROS Assay Kit was also performed as per the manufacturer's instructions, with WT and *NGLY1*-KO (KO-1) HepG2 cells incubating for 48 hours before being trypsinized and counted, with $2.5 \times 10^4$ cells of each line being passaged onto glass bottom 35 mm$^2$ dishes (IWAKI, #3971−035) and subjected to ROS detection 24 hours later as per manufacturer's instructions. Fluorescent microscopy was performed at Ex/Em = 488/517nm using FV3000 confocal microscope (Olympus).

### Iron assay

To determine the $Fe^{2+}$ iron content within cells, an Iron Assay Kit (Sigma Aldrich, #MAK025) was utilized. WT and *NGLY1*-KO (KO-1) HepG2 cells were incubated within DMEM-LG media for 48 hours until trypsinized, collected and counted, before $2.5 \times 10^6$ cells were then passaged for another 24 hours and collected by cell scraper. Samples were then centrifuged at 95$g$ for 5 minutes before re-suspension in provided assay buffer and homogenized by sonication on ice for three gentle pulses at ten seconds each, before being centrifuged again at 1,600$g$ for 10 minutes. The ensuing supernatant was used according to the manufacturer's instructions and absorbance was measured at 593 nm using a Synergy H1 microplate reader.

### Induced Coupling Plasma Mass Spectrometry (ICP-MS)

To gauge the difference in iron content between *NGLY1*-KO (KO-1) and WT HepG2 samples, three biological replicates of each cell line were passaged for 72 hours before being washed by ice cold PBS and collected by cell scraper. Samples were centrifuged for 10 minutes, 215$g$ at 4°C, before the supernatant was aspirated and samples were frozen in liquid nitrogen and then weighed. Quantitative analysis of iron content was performed by ICPMS (Agilent 7700, Agilent Technologies) under support of the RIKEN Core Facilities Management System. Statistical analysis was performed using GraphPad Prism 9 (GraphPad), with a Grubbs Test for Outliers was first performed at 10% Alpha, with $n = 1$ WT sample excluded for being an anomalous value as a result. A one-tailed t-test was then performed with remaining samples with a p-value less than 0.05 considered statistically significant.

### BODIPY™ 581/591 C11 staining analysis via fluorescent microscopy and flow cytometry

To validate the occurrence of lipid peroxidation within *NGLY1*-KO HepG2 cells, BODIPY™ 581/591 C11 (Thermo, #D3861) was used with both fluorescent microscopy and flow cytometry. Fluorescent microcopy was performed by passaging $5 \times 10^4$ cells of WT and *NGLY1*-KO (KO-1) HepG2 cells on glass-bottom 35 $mm^2$ dishes for 72 hours, washed twice with PBS before being stained with 1.5 nM of BODIPY™ 581/591 C11 in DMEM-LG for 20 minutes at 37°C, after which cells were washed with PBS and then visualized at Ex/Em = 581/591nm for lipid staining and Ex/Em = 488/517nm for lipid peroxides using FV3000 confocal microscopy. Flow cytometry was performed by first passaging $3 \times 10^6$ of WT and *NGLY1*-KO (KO-1) HepG2 cells within 100 $mm^2$ petri dishes with DMEM-LG and were allowed to incubate for 72 hours to ~80% confluency, following which they were trypsinised, collected, washed twice with PBS and then stained with 1.5 nM BODIPY™ 581/591 C11 for 20 minutes, after which cells were pelleted at 215$g$ for 5 minutes at 4°C, before samples underwent flow cytometry.

### Ferroptosis inducer treatment of HepG2, HEK293T, and MEF cells

HepG2 cells, HEK293T cells and MEF cells were seeded on a 96-well plate at a density of $1.5 \times 10^4$, $1 \times 10^4$, $5 \times 10^3$ cells/well one day prior to ferroptosis inducer (RSL3 or ML162; Selleck) treatment, respectively. The viability assay was conducted by exposing cells to ferroptosis inducers at concentrations ranging from 0 to 100 µM for 48 h. Following the incubation period, cell viability was determined using CellTiter 96 Aqueous One Solution (MTS) (Promega) in accordance with the manufacturer's protocol.

### Statistical analysis

Statistical analysis was conducted using QIAGEN IPA software and Prism 9. Data were presented as mean ± standard deviation (SD) or standard error of the mean (SEM) as appropriate. Statistical significance was determined using appropriate tests, such as t-test or ANOVA, followed by post-hoc analysis when necessary. A p-value less than 0.05 was considered statistically significant.

## Results

### *NGLY1* deletion in HepG2 cells gave rise to dysregulated ferroptotic and iron metabolism pathways

*NGLY1*-deletion in two *NGLY1*-KO HepG2 cell clones (KO-1 and KO-2) were validated using western blotting (Fig 1A) and genomic sequencing. The sequencing results identified that both clones have the same mutation on exon 1 of the *NGLY1* gene; two nucleotides at the 78−79th position from the start codon (GA) were deleted and resulted in the change of amino acid sequence from MAAAALGSSSGSASPAVAELCQNTPETFLEASKLLLLTYADNIL to MAAAALGSSSGSASPA-VAELCQNTPDLFGGLQAAAHLC* (* is stop codon). The proteomic analysis of WT and *NGLY1*-KO (KO-1) HepG2 cells (Fig 1B) revealed significant alterations in protein quantities. Approximately 700 proteins exhibited significant changes (Fig 1C, red dots), with a higher proportion showing increased compared to decreased levels (p < 0.02) (S1 Table). Notably, our analyses using IPA software identified ferroptosis as a prominently upregulated process in *NGLY1*-KO (KO-1) HepG2 cells (p < 0.02, red dots) (Fig 1D, **Table 1**).

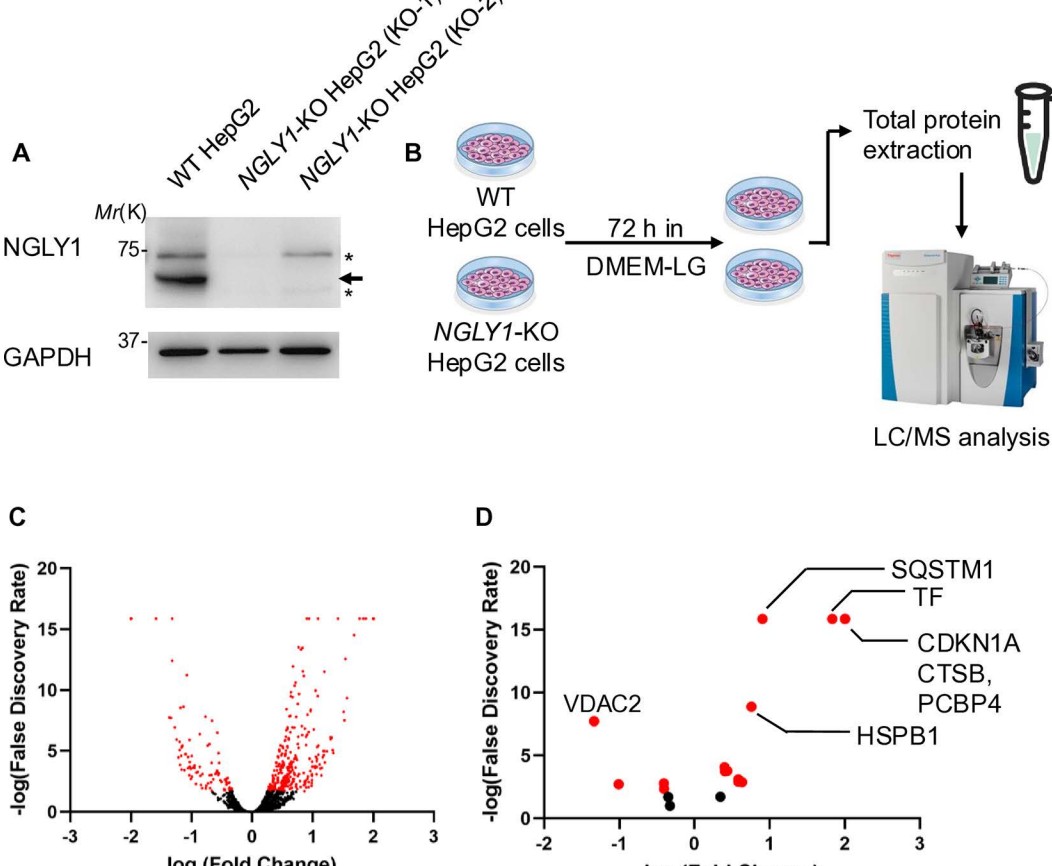

**Fig 1. *NGLY1* deletion produces abnormal protein populations within HepG2 cells.** (A) Cell lysates from WT and *NGLY1*-KO HepG2 cells were subjected to immunoblotting with an antibody against NGLY1. Arrow indicates NGLY1 and asterisks indicate non-specific bands. (B) Flow chart showing the sample preparation from WT and *NGLY1*-KO (KO-1) HepG2 cells for proteomics. (C) Volcano plot depicting the differentially abundant proteins in *NGLY1*-KO (KO-1) compared to WT HepG2 cells. *X*-axis is log2 fold change of *NGLY1*-KO (KO-1) vs WT HepG2 cells and *Y*-axis is the negative logarithm of *p*-value determined by Fisher's Exact Test calculated by IPA software for significance, with dots in red representing p < 0.02 significance. (D) Top significantly dysregulated proteins associated with iron metabolism and ferroptosis in *NGLY1*-KO (KO-1) HepG2 cells with dots in red representing p < 0.02 significance.

**Table 1. Dysregulated canonical pathways in *NGLY1*-KO (KO-1) HepG2 cells.**

| Pathway name | p-value |
|---|---|
| CLEAR (coordinates lysosomal expression and regulation) | 2.31E-06 |
| Ferroptosis | 8.96E-06 |
| FXR/RXR (farnesoid X receptor/retinoid X receptor) activation | 3.03E-05 |
| Germ cell-Sertoli cell junction | 4.26E-05 |
| Xenobiotic metabolism CAR (constitutive androstane receptor) signaling pathway | 5.5E-05 |

TF and TFRC exhibited a substantial increase of approximately 68-fold (**Table 2**, p = 1.36 × 10$^{-16}$) and 2.7-fold (**Table 2**, p = 1.63 × 10$^{-4}$) in *NGLY1*-KO (KO-1) HepG2 cells, respectively. While both TF and TFRC facilitate iron import into the cells [32–34,41], the transferrin receptor 2 (TFR2), a negative regulator of iron-uptake into the cell by regulating the production of the hormone hepcidin [35], was downregulated 10.2-fold (**Table 2**, p = 0.00186). Simultaneously, the presence of the TFRC inhibitor HSPB1 [24] was elevated by 5.7-fold (**Table 2**, p = 1.34 × 10$^{-9}$), indicating an attempt to inhibit TFRC's activity and to reduce cellular iron uptake in an *NGLY1* deleted cellular environment. Other inhibitors of ferroptosis, such as xCT (SLC7A11) [30] and CDKN1A [16], were increased in *NGLY1*-KO (KO-1) HepG2 cells, hinting at cellular attempts to combat ferroptosis (**Table 2**). A notable disturbance of ferroptosis regulatory proteins was also observed, such as a more than 100-fold increase in both ferroptosis promoter cathepsin B (CTSB) [18] and general ferroptosis inhibitor glutamine synthase 2 (GLS2) [42,20] (**Table 2**). However, no significant differential levels in their inducers, STAT3 [43] and p53 [42], respectively, were observed between *NGLY1*-KO (KO-1) and WT samples. Nuclear

**Table 2. Ferroptosis associated proteins in *NGLY1*-KO (KO-1) HepG2 cells. Proteomic analysis of ferroptosis associated protein expression levels within *NGLY1*-KO (KO-1) HepG2 cells compared to WT HepG2 cells (n = 3).**

| Symbol | Function | Expr Fold Change | Expr p-value |
|---|---|---|---|
| CDKN1A | Associated with slower depletion of cellular glutathione and reduced ROS. [16,17] | >100 | 1.36E-16 |
| CTSB | Lysosomal cysteine protease that may play a role in protein turnover. [18] | >100 | 1.36E-16 |
| FDFT1 | Involved in cholesterol biosynthesis. [19] | 2.202 | 0.0189 |
| GLS2 | Catalyzes the synthesis of glutamine from glutamate and ammonia. [20] | >100 | 1.36E-16 |
| H2AC18/H2AC19 | Involved in packaging of DNA into chromatin. [21] | −2.232 | 0.0195 |
| H2AZ2 | Associated with the nucleosome structure in eukaryotes. [22] | >100 | 1.36E-16 |
| H2BC17 | Packages and regulates DNA stability within the nucleus. [23] | −2.545 | 0.00163 |
| HSPB1 | Inhibits iron uptake by acting upon TFRC. [24] | 5.697 | 1.34E-09 |
| NCOA4 | Mediates ferritinophagy. [25,26] | 3.814 | 0.00121 |
| PEBP1 | Mediates interactions between ferroptotic cell death and pro-survival autophagy. [27] | −2.532 | 0.00433 |
| RRAS | Mediate cellular signaling. [28] | 4.28 | 0.00126 |
| SLC39A14 | Reduces iron accumulation in liver cells. [29] | 2.511 | 0.0000833 |
| SLC7A11 | Component of xCT which mediates uptake of cystine that assists GPX4 in mediating ferroptosis. [30] | 3.828 | 0.000775 |
| SQSTM1 | Plays a protective role in protecting cells from stress through p62-Keap1-Nrf2 pathway. [31] | 8.025 | 1.36E-16 |
| TF | Transfers Fe3 + iron into the cell. [32,33] | 67.711 | 1.36E-16 |
| TFRC | Uptakes iron into the cell. [34] | 2.719 | 0.000163 |
| TFR2 | Regulator of iron uptake through hepcidin. [35] | −10.204 | 0.00186 |
| TXNRD1 | Regulates intracellular REDOX environment. [36] | 2.544 | 0.000177 |
| VDAC2 | Voltage-dependent anion-selection channel protein within the mitochondrial membrane. [37] | −21.739 | 1.88E-08 |
| GPX4 | Protects the cells from lipid peroxides by converting them into their respective alcohols. [38–40] | −2.114 | 0.1 |

receptor activator 4 (NCOA4) [25,26] (**Table 2**) was increased 3.8 times in the absence of NGLY1 (p = 0.00121), suggesting a considerable upregulation of ferritinophagy activity. Excessive ferritinophagy has been demonstrated to result in increased release of iron into the cytosol, promoting ferroptosis [44]. Taken together this suggests that the absence of NGLY1 disrupts the homeostasis of iron uptake in HepG2 cells, affecting the inhibitory mechanisms that suppress ferroptosis.

### *NGLY1* deletion has resulted in increased lipid peroxidation and ROS in HepG2 cells

Given the dysregulation of ferroptosis-associated proteins in *NGLY1*-KO HepG2 cells, we next assessed whether this was reflected in cellular lipid peroxidation and oxidative stress, which are the hallmarks of ferroptosis. To investigate the potential lipid peroxidation state, we employed BODIPY™ 581/591 C11 staining, a red-to-green shift fluorescent dye that reports lipid peroxidation in live cells (*i.e.*, a shift toward shorter wavelength emission, from red (595 nm) to green (520 nm), is indicative of the lipid peroxidation level). Microscopy revealed a pronounced increase in green fluorescence in *NGLY1*-KO (KO-1) (Fig 2A) relative to WT HepG2 cells (Fig 2B), indicating enhanced lipid peroxidation. Flow cytometry analysis using BODIPY™ 581/591 C11 supported this observation, showing greater green fluorescence in *NGLY1*-KO (KO-1) cells compared to WT HepG2 cells (Fig 2C), along with increased red fluorescence (Fig 2D). Flow cytometry population mapping revealed that a larger proportion of *NGLY1*-KO (KO-1) than WT HepG2 cells clustered in the lipid peroxides-positive population (Q2) among lipids-positive population (Q1 + Q2) (Fig 2E), whereas WT HepG2 cells exhibited double-positive events (Q2) at a lower frequency (Fig 2F). These results suggested that the elevation of lipid peroxidation in *NGLY1*-KO (KO-1) HepG2 cells was not simply due to the elevation of the lipid content. ROS contributes to the generation of lipid peroxides, therefore, next, we examined intracellular ROS levels. Microscopy demonstrated stronger ROS signals in *NGLY1*-KO (KO-1) HepG2 cells (Fig 2G), particularly along the plasma membrane (yellow arrows), in contrast to the more diffuse staining pattern observed in WT HepG2 cells (Fig 2H). Quantitative ROS assays further confirmed significantly elevated ROS levels in *NGLY1*-KO compared to WT HepG2 cells, while clonal differences in the degree of elevation were noted (Fig 2I). These results collectively show that the loss of NGLY1 leads to increased ROS levels and, as a consequence, increased lipid peroxidation in HepG2 cells.

### *NGLY1* deletion has significant effects on GPX4 levels and iron content

Ferroptosis is a regulated form of cell death driven by iron accumulation and lipid peroxidation [45]. To investigate the effect of NGLY1 loss on iron homeostasis, we first performed proteomic profiling to assess the abundance of iron metabolism–associated proteins. A volcano plot of differentially expressed proteins revealed significant alterations in multiple iron-regulatory proteins in *NGLY1*-KO (KO-1) HepG2 cells compared to WT (Fig 3A, **Table 3**). To validate these findings at the functional level, we quantified total intracellular iron using ICP-MS analysis and observed a significant increase in total iron content ($Fe^{2+}$ and $Fe^{3+}$) in *NGLY1*-KO (KO-1) compared to WT HepG2 cells (Fig 3B). Further speciation analysis showed that this increase was also observed in the bioavailable ferrous ($Fe^{2+}$) iron fraction within *NGLY1*-KO (KO-1) HepG2 cells (Fig 3C). Given that glutathione peroxidase 4 (GPX4) is a key antioxidant enzyme that protects cells from ferroptosis by neutralizing lipid hydroperoxides [69], we next investigated its expression in *NGLY1*-KO and WT HepG2 cells. Interestingly, qPCR analysis revealed a four-fold increase in GPX4 mRNA expression in *NGLY1*-KO (KO-1) HepG2 cells compared to WT HepG2 cells (Fig 3D). In accordance with this transcriptional upregulation, western blot analysis showed a significant increase in GPX4 protein levels in *NGLY1*-KO compared to WT HepG2 cells (Figs 3E–3F). Taken together, these findings suggest that NGLY1 loss leads to increased iron loading and places *NGLY1*-KO HepG2 cells in a hyperactive ferroptotic state, as evidenced by elevated lipid peroxidation and ROS levels. It seems conceivable that the observed upregulation of GPX4 expression in *NGLY1*-KO HepG2 cells may constitute a compensatory response.

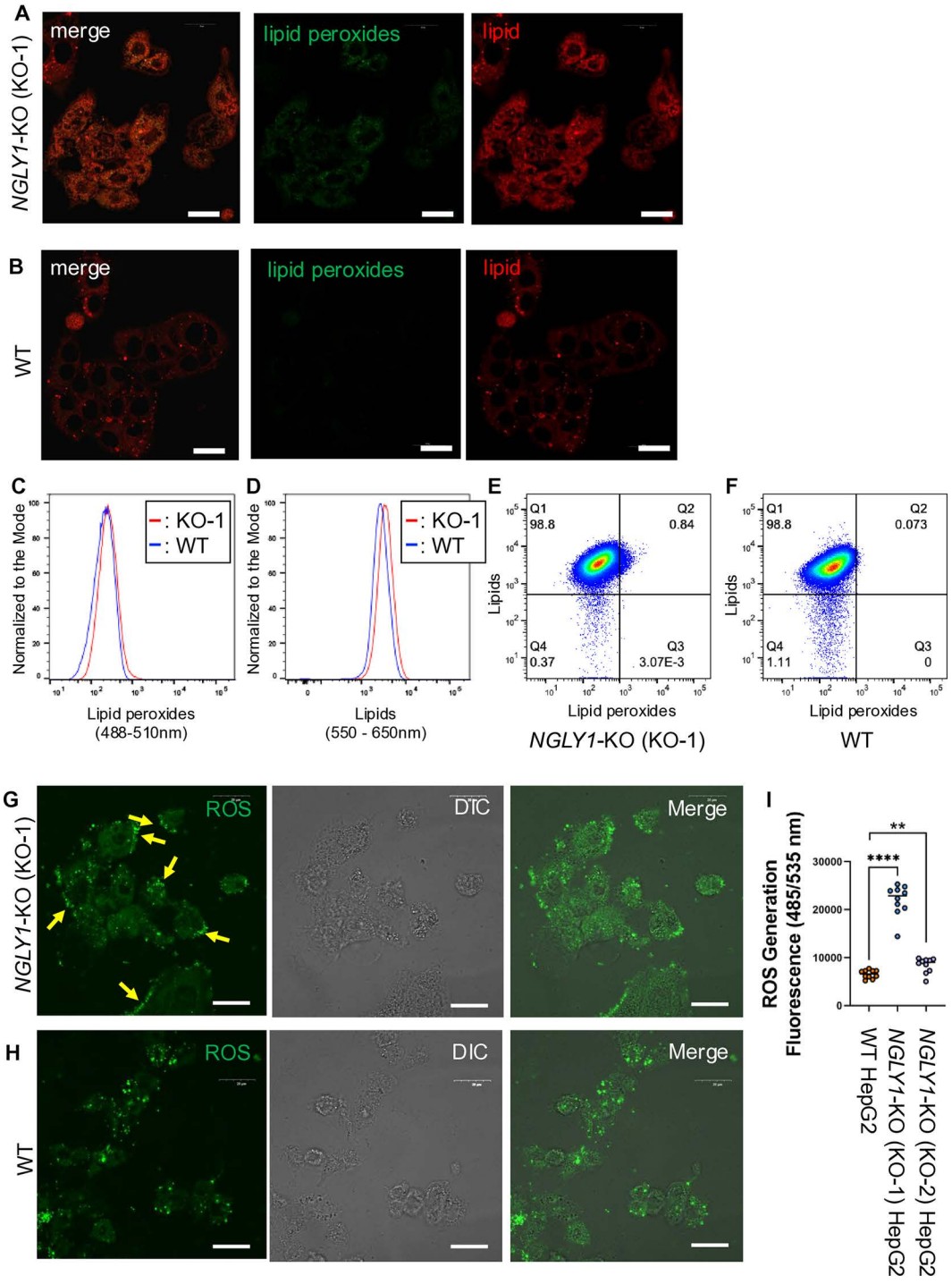

**Fig 2. *NGLY1* deletion leads to increased oxidative stress and lipid peroxidation in HepG2 cells.** (A-B) Fluorescence microscopy of BODIPY™ 581/591 C11–stained HepG2 cells showed increased green fluorescence in *NGLY1*-KO (KO-1) cells (A) relative to WT controls (B), indicating elevated lipid peroxidation (white scalebar: 20 μm). (C-F) Flow cytometry analysis of BODIPY™ 581/591 C11–stained HepG2 cells revealed higher levels of lipid peroxides (C) and lipids (D) in *NGLY1*-KO (KO-1) cells compared to WT (n = 1). Quadrant gating further demonstrated a greater proportion of *NGLY1*-KO (KO-1) cells in the Q2 quadrant (lipid peroxide+/lipid+) (E) relative to WT cells (F). Fluorescence microscopy using the FCFDA/H2DCFDA Cellular ROS kit showed stronger ROS signals in *NGLY1*-KO (KO-1) cells (G), particularly at the plasma membrane (yellow arrows), compared to the diffuse pattern seen in WT cells (H) (white scalebar: 20 μm). (I) Quantification of intracellular ROS levels confirmed significantly elevated ROS in *NGLY1*-KO cells relative to WT (n = 10 per group). (**: p ≤ 0.01, ****: p ≤ 0.0001, all biological replicates unless otherwise indicated; error bars: SEM, A-B and G-H white scalebar: 20 μm).

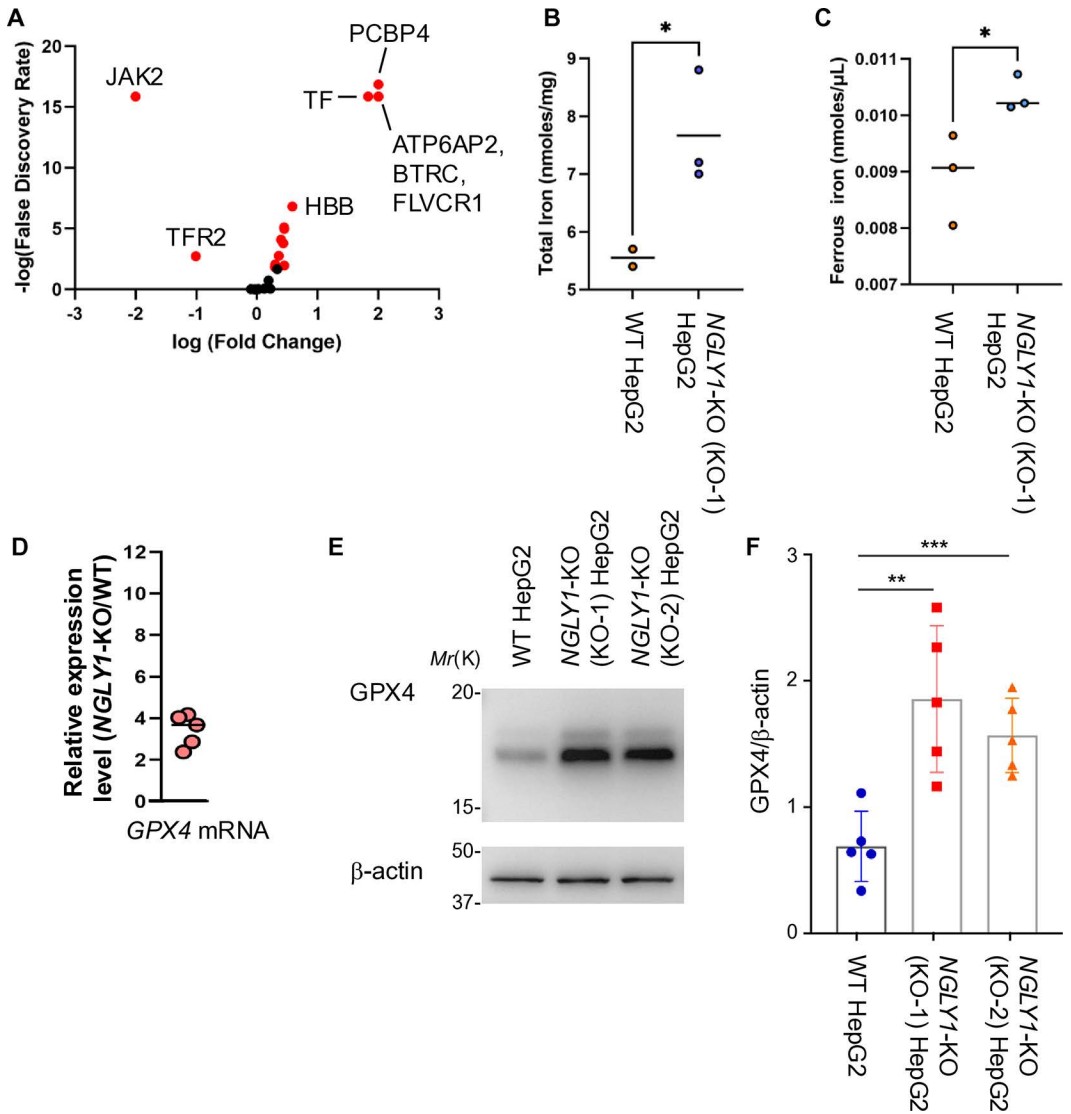

**Fig 3. *NGLY1* deletion alters GPX4 expression and disrupted iron metabolism within HepG2 cells.** (A) Proteomic analysis identified differentially abundant iron metabolism–related proteins in *NGLY1*-KO (KO-1) compared to WT HepG2 cells, visualized in a volcano plot (red dots: p < 0.02, Fisher's Exact Test, IPA software). (B) ICP-MS revealed significantly increased total intracellular iron in *NGLY1*-KO (KO-1) cells compared to WT (KO-1 n = 3, WT n = 2). (C) Bioavailable $Fe^{2+}$ levels were also elevated in *NGLY1*-KO (KO-1) cells relative to WT (n = 3). (D) Relative mRNA expression level of GPX4 mRNA in *NGLY1*-KO (KO-1) HepG2 cells compared to WT controls (n = 5). (E) Western blot analysis exhibited increased GPX4 protein expression in *NGLY1*-KO HepG2 cells compared to WT HepG2 cells (n = 5). (F) Densitometric quantification normalized to β-actin further validated the significant elevation of GPX4 protein in *NGLY1*-KO HepG2 cells. (*: p ≤ 0.05, **: p ≤ 0.01, ***: p ≤ 0.005; error bars: SEM, all biological replicates unless otherwise indicated).

### *NGLY1* deletion significantly increased resistance against ferroptosis in HepG2 cells

To investigate the impact of *NGLY1* deletion on ferroptotic susceptibility, we exposed *NGLY1*-KO and WT HepG2 cells to RSL3 and ML162. Both compounds have been reported to induce ferroptosis through the direct inhibition of GPX4 [70,71]. *NGLY1*-KO HepG2 cells demonstrated enhanced resistance to induced ferroptotic cell death following treatment with RSL3 or ML162 (Figs 4A–4B), in addition to a significant increase in $LD_{50}$ values compared to WT HepG2 cells (Fig 4C–4D). Taken together, this indicates that *NGLY1*-KO HepG2 cells possess an intrinsic resistance to ferroptotic cell

**Table 3. Iron metabolism associated proteins in *NGLY1*-KO (KO-1) HepG2 cells.** Proteomic analysis of iron metabolism associated protein expression levels within *NGLY1-KO* (KO-1) HepG2 cells compared to WT HepG2 cells (n = 3).

| Symbol | Function | Expr Fold Change | Expr p-value |
|---|---|---|---|
| ACSL3 | Mediates accumulation of LDs. [46] | 2 | 0.009 |
| ACSL4 | Contributes to synthesis of PUFA's. [47] | 1.55 | 0.188 |
| HMOX1 | Assists with releasing iron from heme. [48] | 1.323 | 0.86 |
| STEAP3 | Mediates iron metabolism in association with endosomes. [49] | 1.33 | 0.924 |
| DMT1 (CHMP2B) | Iron and non-heme iron transporter. [50] | −1.03 | 0.99 |
| SLC46A1 | Mediator of iron content. [51] | 1.66 | 0.87 |
| FTL | Iron storage within the cell. [52] | −1.092 | 0.99 |
| FTH1 | Iron storage within the cell. [52] | −1.258 | 0.941 |
| RAB7A | Mediates recruitment of LDs. [53] | 1.986 | 0.0153 |
| PCBP1 | Associated with autophagy suppression and iron chaperone. [54] | 1.062 | 0.945 |
| PCBP2 | Iron chaperone. [55] | −1.055 | 0.992 |
| PCBP4 | Possible iron chaperone. [56] | >100 | 1.36E-17 |
| ATG5 | A key pro-autophagy protein that catalyzes ATG8 lipidation. [57] | 1.326 | 0.92 |
| SLC39A14 | Reduces iron accumulation in liver cells. [29] | 2.511 | 0.00008 |
| IREB2 | Assists with regulating iron uptake and storage. [58] | −1.088 | 0.992 |
| ATP6AP1 | H+ Transporter [59] | 2.825 | 0.0109 |
| ATP6AP2 | Possible assembly of the V-ATPase. [60] | >100 | 1.36E-16 |
| ATP6V0A1 | Encodes a1-subunit of the V0 domain of V-ATPases. [61] | 2.301 | 0.00176 |
| ATP6V0D1 | V-ATPase subunit. [62] | 2.83 | 0.0000104 |
| BTRC | Mediates ubiquitination and degradation of target proteins. [63] | >100 | 1.36E-16 |
| EGFR | Regulate cell proliferation, survival, differentiation and migration. [64] | 2.162 | 0.0221 |
| FLVCR1 | Regulates intracellular heme accumulation. [65] | >100 | 1.36E-16 |
| HBA1/HBA2 | The alpha chains of hemoglobin. [66] | 2.805 | 0.00000782 |
| HBB | The beta chains of hemoglobin. [66] | 3.833 | 1.49E-07 |
| JAK2 | Tyrosine Kinase involved in cellular growth and proliferation. [67,68] | −100> | 1.36E-16 |
| TF | Transfers Fe3+iron into the cell. [32,33] | 67.711 | 1.36E-16 |
| TFRC | Uptakes iron into the cell. [34] | 2.719 | 0.000163 |
| TFR2 | Regulator of iron uptake through hepcidin. [35] | −10.204 | 0.00186 |

death. This resistance is consistent with our observations of increased GPX4 expression in *NGLY1*-KO cells (Figs 3D–3F), which is known to inhibit ferroptosis and may contribute to the observed protective phenotype. Earlier reports indicated the heightened sensitivity to ferroptotic cell death of *NGLY1*-KO A549 cells as well as their diminished GPX4 protein levels [7]. Therefore, we conducted the same experiments using cell lines other than HepG2 cells to examine the context-dependency of the impact of *NGLY1* loss on ferroptotic susceptibility. Our results, obtained by using HEK293T and MEF cells, showed that *NGLY1*-KO HEK293T cells exhibit increased resistance to induced ferroptotic cell death, accompanied by elevated GPX4 protein levels when compared with WT HEK293T cells (Figs 4E, 4G, 4I). Conversely, *Ngly1*-KO MEF cells exhibited heightened sensitivity to induced ferroptotic cell death, characterized by reduced GPX4 protein levels in comparison to WT MEF cells (Figs 4F, 4H, 4J). Taken together, these results highlight the context dependency of induced ferroptotic cell death resistance and GPX4 expression levels on *NGLY1*-KO/*Ngly1*-KO cell type.

## Discussion

NGLY1 deficiency is a rare but devastating autosomal recessive disorder whose mechanisms remain poorly understood. In this study, we investigated the role of NGLY1 in modulating ferroptotic and iron metabolism pathways in HepG2 cells,

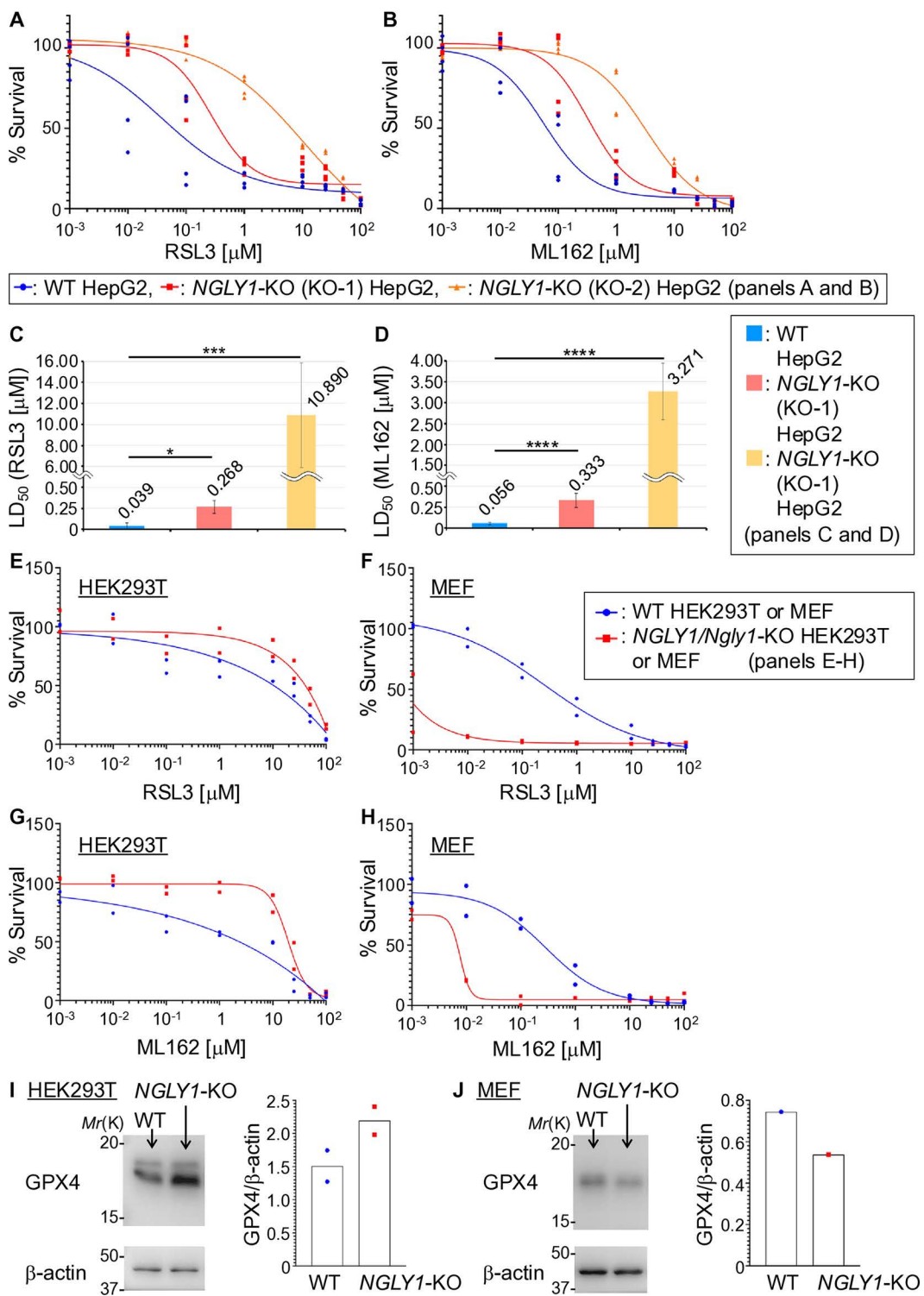

**Fig 4. *NGLY1* deletion conferred resistance to ferroptosis-induced cell death in HepG2 cells.** (A-B) *NGLY1*-KO (red: KO-1, orange: KO-2) HepG2 cells showed significantly greater survival compared to WT (blue) following RSL3 (A) or ML162 (B) treatment. (C-D) Median lethal dose (LD$_{50}$) of RSL3 (C) and ML162 (D) of *NGLY1*-KO (KO-1 and KO-2) and WT HepG2 cells (n = 4). (E, G) Cell death in WT and *NGLY1*-KO HEK293T cells treated with RSL3 (E) or ML162 (G) (n = 2). (F, H) Cell death in WT and *Ngly1*-KO mouse embryonic fibroblast (MEF) cells treated with RSL3 (F) or ML162 (H) (n = 2). (I-J) Western blot analysis using cell lysates from WT and *NGLY1*-KO/*Ngly1*-KO HEK293T cells (I, n = 2) or MEF cells (J, n = 1). Bar graphs in panels I-J show the results of densitometric quantification. (\*\*: p ≤ 0.01, \*\*\*: p ≤ 0.005 \*\*\*\*: p ≤ 0.0001, error bars: SEM; biological replicates unless otherwise noted).

using a combination of proteomic, genetic, and biochemical analyses. Our results indicate that NGLY1 absence in HepG2 cells results in significant dysregulation of iron metabolism and the processes associated with ferroptosis. Proteomic analysis revealed a marked increase in proteins associated with ferroptosis and iron uptake, notably TF and TFRC respectively, alongside a downregulation of regulatory elements such as TFRC2 and inhibitors like HSPB1. Quantification of cellular iron uptake revealed a notable elevation in the levels of total and ferrous iron in the *NGLY1*-KO (KO-1) HepG2 cells, including a disruption of cellular iron homeostasis. Additionally, fluorescence microscopy and flow cytometry demonstrated that *NGLY1*-KO (KO-1) HepG2 cells exhibited elevated levels of lipid peroxidation and ROS relative to WT cells. The observed increase in GPX4 mRNA transcription in *NGLY1*-KO (KO-1) HepG2 cells, concomitant with increased GPX4 protein levels in *NGLY1*-KO HepG2 cells, was observed (Figs 3D–3F). Together, these findings suggest that ferroptotic susceptibility and dysregulation of iron metabolism emerge in hepatic cells following *NGLY1* deletion, while concomitantly increasing GPX4 expression as a remedial measure. *NGLY1*-KO HepG2 cells showed a higher resistance to induced ferroptotic cell death compared to WT HepG2 cells, though clonal differences regarding the extent of resistance were observed (Fig 4). Both ferroptosis inducers utilized in this study, RSL3 and ML162, have been shown to target GPX4. However, the elevated expression of GPX4 in *NGLY1*-KO HepG2 cells results in an increased resistance to the induced ferroptotic cell death. These findings further support the notion that elevated GPX4 expression exerts a protective function in counteracting persistent ferroptotic stress in *NGLY1*-KO HepG2 cells.

The role of TF/TFRC in iron intake is well documented [32–34]. $Fe^{3+}$ is bound to TF and enters the cell, via the endocytic pathway. In the endosomal compartment the acidic environment induces a conformational change in TF, resulting in the release of $Fe^{3+}$ [32,33]. During this process, $Fe^{3+}$ is reduced to $Fe^{2+}$ (Fig 5). Subsequently, $Fe^{2+}$ is transported out of

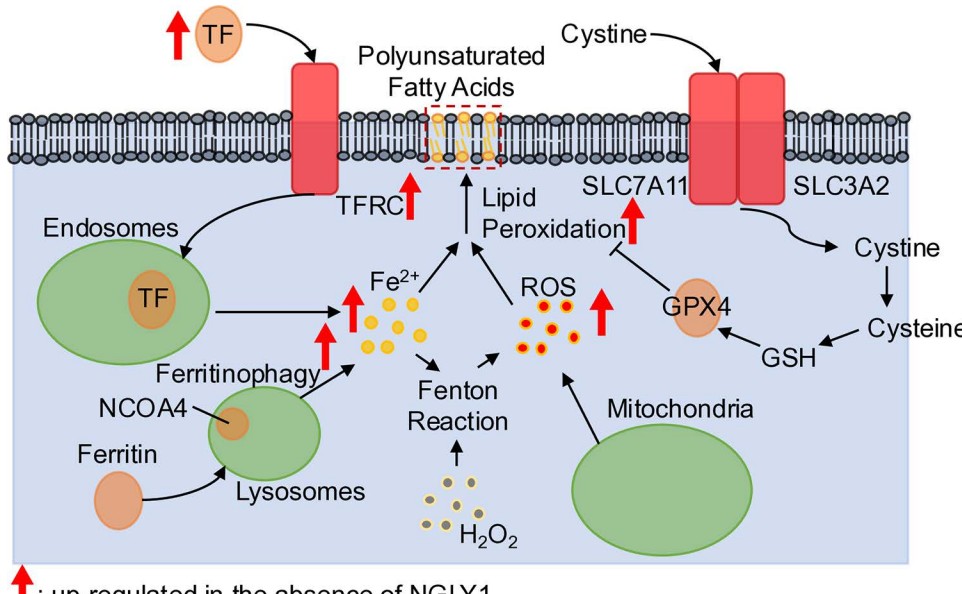

: up-regulated in the absence of NGLY1

**Fig 5. Excessive iron content within the cellular cytoplasm produces ferroptosis by peroxidation of polyunsaturated fatty acids.** Schematic representation of the link between iron metabolism, ROS generation and ferroptosis. TF is shuttled into the cytoplasm by TFRC, where it is taken into endosomes and releases its iron cargo, which will release as $Fe^{2+}$ into the cytoplasm that can interact with PUFAs in the cellular membranes and create lipid peroxides, in addition to interacting with free $H_2O_2$ to produce ROS by the fenton reaction [72,73]. Simultaneously, ferritin and NCOA4 can be taken into lysosomes and undergo ferritinophagy, which releases additional $Fe^{2+}$ into the cytoplasm. Meanwhile, dysregulated mitochondria can release ROS into the cytoplasm which contributes to lipid oxidation of lipids. As a counter process, cystine enters the cell via the xCT complex (SLC7A11/SLC3A2) where it is hydrolyzed into cysteine, which, along with glutamate, is synthesized into GSH which acts as a substrate for GPX4 to convert into GSSG to reduce lipid peroxides into corresponding alcohols, inhibiting ferroptosis.

the endosome and into the cytosol by divalent metal transporter 1 (DMT1). The proteomics data indicate that DMT1 levels remain relatively unaltered between *NGLY1*-KO (KO-1) and WT samples (**Table 3**). However, SLC39A14 solute carrier performs a similar function and was observed to have 2.5-fold increase in *NGLY1*-KO (KO-1) compared to WT samples (**Table 3**, $p = 8 \times 10^{-5}$). To date, no direct link between NGLY1 and TF/TFRC has been established, though both TF and TFRC have been reported to be *N*-glycosylated [74,75]. Further studies are required to elucidate whether these proteins are indeed NGLY1 substrates and to determine the precise manner in which NGLY1 affects the levels of TF/TFRC and iron metabolism in the liver.

The combination of unbound $Fe^{2+}$ and hydrogen peroxide ($H_2O_2$) in the cytoplasm significantly amplifies cellular ROS levels via the fenton reaction [72,73]. Iron content and ROS levels were significantly elevated in *NGLY1*-KO compared to WT HepG2 cells (Figs 3B and 2I, respectively). Elevated $Fe^{2+}$ and ROS levels promote lipid peroxidation of poly-unsaturated fatty acyl chains (PUFA) in the cell membranes [11,12,76], a significant driving force behind ferroptosis. A more localized ROS staining was observed in *NGLY1*-KO (KO-1) cells (Fig 2G), which was found to be associated with cellular membranes, in comparison to the more diffused ROS staining observed in WT HepG2 cells (Fig 2H). To compensate for heightened ROS driven lipid peroxidation, *NGLY1*-KO (KO-1) HepG2 cells significantly upregulated GLS2 (**Table 2**, $p = 1.36 \times 10^{-16}$) by a factor of more than 100. GLS2 plays a crucial role in the glutathione (GSH) biosynthesis by catalyzing the conversion of glutamine to glutamate. GPX4, in turn, prevents the formation of lipid hydroperoxides and maintains cellular redox homeostasis by oxidizing GSH. qPCR results showed increased mRNA levels of GPX4 in *NGLY1*-KO (KO-1) compared to WT HepG2 cells (Fig 3D). Concomitantly, GPX4 protein levels exhibited significant increases in *NGLY1*-KO HepG2 cells (Figs 3E–3F), likely enhancing the cells ability to endure the consequences of elevated ROS-driven lipid peroxidation (Figs 2A, 2C, 2E and 2I). The protein levels of ferritin, the primary iron storage protein composed of ferritin heavy chain (FTH1) and ferritin light chain (FTL), remained unchanged (**Table 3**), while NCOA4 expression was increased. NCOA4 expression promotes ferritinophagy, a selective autophagic process that degrades ferritin in the lysosomal system, thereby reducing the cell's iron storage capacity and releasing further ROS into the cellular environment. Taken together, these results suggest that ferroptosis is a chronic phenomenon in *NGLY1*-KO HepG2 cells. In the presence of ferroptosis inducers RSL3 and ML162 [70,71], WT HepG2 cells exhibited significantly higher sensitivity towards both compounds in comparison to *NGLY1*-KO HepG2 cells (Figs 4A–4D). This suggests that *NGLY1*-KO HepG2 cells may exist in a partially adapted, ferroptosis-tolerant state which has been bolstered by increased GPX4 levels (Figs 3E–3F). The full extent of this persistent ferroptotic state and its impact on cellular viability remain open questions, representing potential avenues for future scientific investigation.

Regarding the context dependency of GPX4 expression levels on *NGLY1*-KO cell type, the public database "NGLY1 browser (https://apps.embl.de/ngly1browser/)" reports that NGLY1 deficiency patient-derived fibroblasts show increased GPX4 mRNA and reduced GPX4 protein levels, while patient-derived lymphoblastoid cells showed decreased GPX4 mRNA and protein levels. This observation also highlights the context dependency of GPX4 mRNA and protein expression levels on *NGLY1*-KO cell type.

The absence of NGLY1 has previously been associated with ferroptosis through the NGLY1-NRF1 pathway [7]. Nevertheless, NRF1 was not detected in our proteomic analysis, which is believed to be the consequence of its constitutive degradation. Nevertheless, some of the proteins whose expression is regulated by NRF1 were up- or down-regulated in *NGLY1*-KO HepG2 cells (*e.g.,* up-regulated: GIT1, BTRC, PAK1, GCLM; down-regulated: MIS12) (S1 Table). In addition to NRF1, the genes in question can also be regulated by nuclear factor erythroid 2-related factor 2 (NRF2) [77]. Given this partial overlap in gene regulation, it remains challenging to attribute the observed increased oxidative "pro-ferroptotic" state and the compensatory upregulation of GPX4 to tolerate this state in our *NGLY1*-KO HepG2 cells to a specific nuclear transcription factor, highlighting the need for further studies.

In conclusion, the loss of *NGLY1* gives rise to a dysregulated iron metabolism and increased ROS levels despite the cellular antioxidant response, as evidenced by elevated GPX4 level. The interplay between NGLY1, ferroptosis, and iron

metabolism highlights the significance of NGLY1 in maintaining cellular homeostasis and its potential implications for human health, particularly in the context of liver-related disorders and diseases involving iron dysregulation and oxidative stress. It seems noteworthy, that despite their chronic ferroptotic state, *NGLY1*-KO HepG2 cells continue to demonstrate growth and division. This paradoxical survival, despite the initiation of programmed cell death, suggests a 'sublethal' engagement of ferroptosis, where the cells endure sustained oxidative and iron-related stress without fully committing to cell death. Alternatively, the enhanced expression of ferroptosis-resistance markers such as GPX4 may raise the threshold required to trigger ferroptotic cell death. This regulated or partial ferroptotic state might allow the cells to balance between survival and stress adaptation, raising questions about the threshold conditions for ferroptotic lethality and the potential for adaptive responses within this pathway. Further research is warranted to understand the precise molecular mechanisms underlying these observations, and to explore potential therapeutic interventions targeting NGLY1-related pathways.

## Supporting information

**S1 Fig. (A-B) Uncropped images of blots shown in figures (A: Fig 1A, 1B: Fig 3E, 3C: Figs 4I-4J).**
(PDF)

**S1 Table. Up- and down-regulated proteins in *NGLY1*-KO (KO-1) HepG2 cells.**
(XLSX)

## Acknowledgments

We wish to thank the members of Glycometabolic Biochemistry Laboratory (RIKEN).

## Author contributions

**Conceptualization:** Stuart Emmerson, Haruhiko Fujihira, Naoshi Dohmae, Tadashi Suzuki.

**Data curation:** Stuart Emmerson, Haruhiko Fujihira, Takehiro Suzuki, Tadashi Suzuki.

**Formal analysis:** Stuart Emmerson, Haruhiko Fujihira, Takehiro Suzuki, Naoshi Dohmae, Tadashi Suzuki.

**Funding acquisition:** Haruhiko Fujihira, Tadashi Suzuki.

**Investigation:** Stuart Emmerson, Haruhiko Fujihira, Takehiro Suzuki.

**Methodology:** Takehiro Suzuki.

**Supervision:** Haruhiko Fujihira.

**Validation:** Stuart Emmerson, Haruhiko Fujihira, Takehiro Suzuki, Naoshi Dohmae.

**Visualization:** Stuart Emmerson.

**Writing – original draft:** Stuart Emmerson, Haruhiko Fujihira, Takehiro Suzuki, Tadashi Suzuki.

**Writing – review & editing:** Stuart Emmerson, Haruhiko Fujihira, Takehiro Suzuki, Naoshi Dohmae, Peter Greimel, Yoshio Hirabayashi, Tadashi Suzuki.

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
