## [Decision Letter · Decision Letter 0]

22 Jan 2025

Dear Dr. Fujihira,

Thank you for submitting your manuscript to PLOS ONE. After careful consideration, we feel that it has merit but does not fully meet PLOS ONE’s publication criteria as it currently stands. Therefore, we invite you to submit a revised version of the manuscript that addresses the points raised during the review process.

**ACADEMIC EDITOR: **

Explore mechanism for GPX4 downregulationPerform rescue experiments with exogenous GPX4 or another key factor downregulated in NGLY1 KODirectly assess whether NGLY1 loss in HepG2 cells affects sensitivity to induction of ferroptosis

publication criteria  and not, for example, on novelty or perceived impact.

We look forward to receiving your revised manuscript.

Kind regards,

Kostas Pantopoulos, PhD

Academic Editor

PLOS ONE

2. Thank you for stating the following financial disclosure:  [This work was supported by RIKEN Pioneering Project (“Glyco-Lipidologue Initiative”) (to TS), Japan Agency for Medical Research and Development-Core Research for Evolutional Science and Technology (AMED-CREST, JP24gm14100003) (to TS and HF).].  Please state what role the funders took in the study.  If the funders had no role, please state: "The funders had no role in study design, data collection and analysis, decision to publish, or preparation of the manuscript." If this statement is not correct you must amend it as needed.

Additional Editor Comments (if provided):

Reviewers' comments:

Reviewer's Responses to Questions

**Comments to the Author**

1. Is the manuscript technically sound, and do the data support the conclusions?

Reviewer #1: Yes

Reviewer #2: Yes

2. Has the statistical analysis been performed appropriately and rigorously?

Reviewer #1: Yes

Reviewer #2: Yes

3. Have the authors made all data underlying the findings in their manuscript fully available?

Reviewer #1: Yes

Reviewer #2: Yes

4. Is the manuscript presented in an intelligible fashion and written in standard English?

Reviewer #1: Yes

Reviewer #2: Yes

Reviewer #1: In this manuscript, Emmerson et al. report that cytosolic N-glycanase (NGLY1) is a regulator of iron homeostasis and ferroptosis in hepatoma cells. The authors found that the expression levels of several key factors for ion metabolism and related oxidative stress in NGLY1 KO HepG2 cells. Concomitantly, they demonstrated that the KO cells display the increased levels of ferrous, ROS and peroxidized lipids. Overall, although the mechanistic investigation was not fully performed, the conclusions of the paper are clear and all the methods are scientifically sound. I am happy to support publication of the paper after appropriate revision. Specific points are as follows.

Major points

1. The mechanism of how GPX4 protein is downregulated is not examined. To explain this phenomenon in NGLY1 KO cells, any logical hypothesis or experimental evidence should be provided.

2. The importance of the alterations of the key factors shown should be provided more. For instance, rescue experiments by exogenous expression of GPX4 or another key factor downregulated in NGLY1 KO would be one choice.

3. Cell type specificity is not discussed or examined. Analysis of previous or public proteomic or transcriptomic data from systemic KO mice or NGLY1-deficient patient-derived cells could provide a new clue.

4. Fig. 2G. The authors showed the increase in neutral lipids in KO cells. Please provide more detailed and reasonable explanation of the Fig. 2F and G data.

Minor points

5. Line 30-31. Structure of this sentence needs to be revised. “was” meant to be “were”?

6. Methods. Regarding centrifugation conditions, only “rpm” but not “g” is written in some places. Please define with g or rpm with the rotor name(s).

7. L216. “sequence confirmation” needs to be revised with more detailed description. For instance, genomic sequencing.

8. Fig. 2C. The aspect ratio of the graph seems strange and should be corrected.

9. Typo

Line 102. Space is needed between value and unit, “1.0g/L”.

Line 121. NP-40 should be spelled out when it first appears.

Line 123. Space is needed for “pH8.0”.

Line 257. Space is missing for 595nm and 520nm.

Line 333. “are” should be “is”?

Reviewer #2: The authors present an analysis of the consequences of loss of N-glycanase (encoded by the gene NGLY1) on cellular metabolism, with a particular focus on pathways implicated in oxidative stress and iron metabolism, using a human hepatocellular carcinoma cell line HepG2 cells. Loss of NGLY1 function is observed in the rare human genetic disorder NGLY1 Deficiency, characterized by global developmental delay, a hyperkinetic movement disorder, transient elevation of liver transaminases, peripheral neuropathy, and other features. The goal of this study is to probe the molecular mechanisms by which loss of N-glycanase drives these diverse features.

The manuscript is clear and presents some novel findings, notably reporting on disturbances in iron metabolism in NGLY1 deficient cells. However, my only real criticism is that the study represents a very modest increase in knowledge about N-glycanase’s role in regulating cellular metabolism and pathways involved in disease pathogenesis.

Summary – the manuscript presents a proteomic analysis of changes in protein levels in NGLY1 deficient HepG2 cells, noting a pronounced disturbance in pathways implicated in iron metabolism and ferroptosis, an iron-dependent oxidative form of cell death. The authors characterize changes in protein and mRNA levels associated with iron handling and ferroptosis, observe elevated iron influx, changes in key transferrin pathway members, increases in reactive oxygen species and lipid peroxidation.

Criticisms:

The manuscript details changes in markers/proteins/genes associated with ferroptosis regulation, but does not directly measure whether NGLY1 loss in HepG2 cells affects sensitivity to induction of ferroptosis (a form of cell death) as was reported earlier by Forcina et al (2022) https://doi.org/10.1073/pnas.2118646119. As result, statements such as “ferroptosis is prominently upregulated” (abstract), cells are in a “perpetual state of ferroptosis” (p. 4, 14) are misleading. More accurate would be to state that pathways implicated in ferroptosis regulation are altered. This is addressed somewhat in the discussion section, in which the authors note that the mutant cells are capable of cell growth and division, but may show “a ‘sublethal’ engagement of ferroptosis”. A better way to resolve this would be to measure whether the cells are in fact more sensitive to ferroptosis induction by, for example, GPX4 inhibitors like RSL3 or ML 162 as was shown by Forcina et al.

The work is entirely conducted using a single cell line HepG2. (The section title in line 264 “NGLY1 deletion increased iron content in human liver cells” should refer to a “human liver cell line” rather than “human liver cells”). The findings would be substantially extended if similar changes were observed in primary liver cells, for example from NGLY1 deficient rodent models. Given that the data is based on mutant clones of an immortal cell line selected for the ability to grow (and thus has acquired compensatory mechanisms to survive increased ferroptotic/oxidative stress), there is reason to be concerned that phenotypic changes may be very different in primary cells.

Minor criticisms:

- There are substantial differences in the levels of ROS generation between KO lines 1 and 2 (Figure 3D). Is there any explanation for that difference? Nonetheless, the authors should be encouraged to note that difference, even if it does not change the conclusion of that analysis.

- The figure legend to Figure 3E should provide an explanation for the yellow arrows (currently in the results section text)

- Line 332: “believed to be the consequence of its perineal degradation” I believe the authors mean to say “persistent”, “constant”, or “constitutive” degradation.

**Do you want your identity to be public for this peer review?** For information about this choice, including consent withdrawal, please see our Privacy Policy

Reviewer #1: **Yes: ** Yasuhiko Kizuka

Reviewer #2: **Yes: ** Kevin Lee

---

## [Author Response · Author response to Decision Letter 1]

30 May 2025

We wish to extend our profound gratitude to the reviewers for their constructive comments. In light of the reviewer’s comments, a number of supplementary experiments were conducted. During the course of these new experiments, we were not able to reproduce our originally reported decrease in GPX4 protein levels in NGLY1-KO HepG2 cells compared to WT HepG2 cells. In order to ensure the highest possible degree of reproducibility, the recent data set was reproduced independently by two researchers, yielding consistent results. This recent data set revealed no reduction in GPX4 protein levels in NGLY1-KO HepG2 cells under any of the tested experimental conditions. In contrast, this recent data set displayed an increase in the GPX4 protein levels in NGLY1-KO HepG2 cells in comparison with WT HepG2 cells, within the specified experimental parameters. These recent findings are more consistently aligned with the previously stated increase in GPX4 mRNA expression levels as outlined in the original submission.

Further experiments were conducted to assess the sensitivity of NGLY1-KO and WT HepG2 cells to ferroptosis induction by GPX4 inhibitors (RSL3 and ML162). NGLY1-KO HepG2 cells exhibited increased resistance to induced ferroptotic cell death in comparison to WT HepG2 cells. These findings are consistent with previous reports associating elevated GPX4 levels with increased resistance to induced ferroptotic cell death, thereby providing additional support for our present finding of increased GPX4 protein levels in NGLY1-KO HepG2 cells. In light of this aforementioned consistency of our present findings concerning elevated GPX4 protein levels in NGLY1-KO HepG2 cells, we have revised our manuscript to reflect our recent data set, as detailed below.

Revisions to Figures

1. Fig. 2: previous Figs. 2A-B were removed; Fig. 2C was moved to Fig. 3D; previous Figs. 3D-F were moved to Figs. 2G-I; new panels were added as Figs. 2E-F showing population mappings of flow cytometry

2. Fig. 3; as stated above, Figs. 3D-F were moved to Fig. 2G-I; previous Fig. 2C was moved to Fig. 3D; newly obtained GPX4 western blot results were added as Figs. 3E-F.

3. Fig. 5; previous Fig. 4 was moved to Fig. 5.

4. Fig. 4; new figures were added showing the present data set of ferroptosis sensitivity assays.

Revisions to Supplemental Figures

1. Fig. S2; Fig. S1 was moved to Fig. S2.

2. Fig. S1; new figures were added showing the present data set of ferroptosis sensitivity assays and western blotting of GPX4 using HEK293T and MEF cells.

*In accordance with the above outlined revisions, all associated Figure references in the text and associated Figure legends were revised.

Revisions to “Results” section

1. The new results were outlined in the second and third subsection of the ‘Results’ section. The titles of the two sections were revised from “NGLY1 deletion had significant effects on GPX4 levels” and “NGLY1 deletion increased iron content in human liver cells” to “NGLY1 deletion has resulted in increased lipid peroxidation and ROS in HepG2 cells” and “NGLY1 deletion has significant effects on GPX4 levels and iron content” (pages 10-12, lines 250-290).

2. A fourth subsection titled ‘NGLY1 deletion significantly increased resistance against ferroptosis in HepG2 cells’ was included to explain the new findings based on the ferroptosis induction experiments (pages 12, line 291-300). To better reflect these recent findings, the end of the last sentence of the first subsection of the ‘Results’ section (page 10, line 249) was changed from “impairing the inhibitory mechanism that suppress ferroptosis” to “affecting the inhibitory mechanism that suppress ferroptosis”.

Revisions to “Discussion” section

1. page 13, lines 311-312: To better reflect our recent findings we removed the sentence “Notwithstanding the observed upregulation of GPX4 mRNA, the levels of GPX4 protein were significantly reduced in NGLY1-KO HepG2 cells, indicating that GPX4 is subjected to post-translational down-regulations.”. And changed the beginning of the following sentence from “Fluorescence microscopy~” to “Additionally, fluorescence microscopy~”.

2. page 13, lines 313-315: The following sentence was inserted “The observed increase in GPX4 mRNA transcription in NGLY1-KO (KO-1) HepG2 cells, concomitant with increased GPX4 protein levels in NGLY1-KO HepG2 cells, was observed (Figs. 3D-F).”.

3. page 13, line 315: The start of the sentence was changed from “These findings suggest~” to “Together, these findings suggest~”.

4. page 13, lines 317-323: The final sub-sentence of the last line in the first paragraph of the ‘Discussion’ section was changed from “following NGLY1 deletion, offering a novel perspective on the molecular mechanisms underlying iron-related oxidative stress.” to “, while concomitantly increasing GPX4 expression as a remedial measure. NGLY1-KO HepG2 cells showed a higher resistance to induced ferroptotic cell death compared to WT HepG2 cells, though clonal differences regarding the extent of resistance were observed (Fig. 4). Both ferroptosis inducers utilized in this study, RSL3 and ML162, have been shown to target GPX4. However, the elevated expression of GPX4 in NGLY1-KO HepG2 cells results in an increased resistance to the induced ferroptotic cell death. These findings further support the notion that elevated GPX4 expression exerts a protective function in counteracting persistent ferroptotic stress in NGLY1-KO HepG2 cells”.

5. page 14, lines 347-349: The text passage “However, GPX4 protein levels exhibited a significant decline in NGLY1-KO (both KO-1 and KO-2) cells (Figs. 2A-B), thereby constraining their capacity to neutralize ROS-driven lipid peroxidation. It remains to be determined whether the GPX4 protein downregulation in NGLY1-KO HepG2 cells is the result of a reduction in protein translation, disruption of post-translational modification or reduced protein stability” was revised to “Concomitantly, GPX4 protein levels exhibited significant increases in NGLY1-KO HepG2 cells (Figs. 3E-F), likely enhancing the cells ability to endure the consequences of elevated ROS-driven lipid peroxidation (Figs. 2A, 2C, 2E and 2I).”.

6. page 14, lines 355-358: The following text passage was added: “In the presence of ferroptosis inducers RSL3 and ML162 (32, 33), WT HepG2 cells exhibited significantly higher sensitivity towards both compounds in comparison to NGLY1-KO HepG2 cells (Figs. 4A-D). This suggests that NGLY1-KO HepG2 cells may exist in a partially adapted, ferroptosis-tolerant state which has been bolstered by increased GPX4 levels (Figs. 3E-F).”

7. pages 14-15, lines 361-371: An additional paragraph discussing the new findings was inserted, stating: “Earlier reports indicated the heightened sensitivity to ferroptotic cell death of NGLY1-KO A549 cells as well as their diminished GPX4 protein levels (7). The public database “NGLY1 browser (https://apps.embl.de/ngly1browser/)” reports that NGLY1 deficiency patient-derived fibroblasts show increased GPX4 mRNA and reduced GPX4 protein levels, while patient-derived lymphoblastoid cells showed decreased GPX4 mRNA and protein levels. Moreover, preliminary results indicate that NGLY1-KO HEK293T cells exhibit increased resistance to induced ferroptotic cell death, accompanied by elevated GPX4 protein levels when compared with WT HEK293T cells. Conversely, Ngly1-KO MEF cells exhibited heightened sensitivity to induced ferroptotic cell death, characterized by reduced GPX4 protein levels in comparison to WT MEF cells (Figs. S1A-F). Taken together, these results highlight the context dependency of induced ferroptotic cell death resistance and GPX4 mRNA and protein expression levels on NGLY1-KO cell type.”.

8. page 15, line 382: “GPX4 mRNA level” was changed to “GPX4 level”.

9. page 16, line 389: A sentence was added, and the beginning of the following sentence was adjusted to improve flow as follows: “Such a regulated or partial ferroptotic~” changed to “Alternatively, the enhanced expression of ferroptosis-resistance markers such as GPX4 may raise the threshold required to trigger ferroptotic cell death. This regulated or partial ferroptotic~”. In the same sentence the words “stress adaption” were changed to “stress adaptation”.

Revisions to “Methods” section

1. The method for “Ferroptosis inducer treatment of HepG2 cells” was added to the methods section (page 9, lines 207-212).

In the following section a point-by-point response to the reviewers’ comments is provided.

Reviewer #1:

Comments:

In this manuscript, Emmerson et al. report that cytosolic N-glycanase (NGLY1) is a regulator of iron homeostasis and ferroptosis in hepatoma cells. The authors found that the expression levels of several key factors for ion metabolism and related oxidative stress in NGLY1 KO HepG2 cells. Concomitantly, they demonstrated that the KO cells display the increased levels of ferrous, ROS and peroxidized lipids. Overall, although the mechanistic investigation was not fully performed, the conclusions of the paper are clear and all the methods are scientifically sound. I am happy to support publication of the paper after appropriate revision. Specific points are as follows.

Major points

1. The mechanism of how GPX4 protein is downregulated is not examined. To explain this phenomenon in NGLY1 KO cells, any logical hypothesis or experimental evidence should be provided.

>As outlined above in more detail, two independent researchers confirmed the upregulation of GPX4 protein levels in NGLY1-KO HepG2 cells. This highly reproducible increase in GPX4 protein levels is now consistent with the reported increase in GPX4 mRNA levels. The new experimental results are shown in Figs. 3E-F.

2. The importance of the alterations of the key factors shown should be provided more. For instance, rescue experiments by exogenous expression of GPX4 or another key factor downregulated in NGLY1 KO would be one choice.

>To assess this point, we treated the cells with GPX4 inhibitors (RSL3 and ML162) and checked their sensitivity to inhibitor induced ferroptotic cell death. The results are shown in new Figs. 4A-D. The high GPX4 expression resulted in a high resistance to the induced ferroptotic cell death. This indicates that the alteration of GPX4 levels in NGLY1-KO HepG2 cells plays an important role in the acquisition of ferroptotic cell death resistance compared to WT HepG2 cells. The same relationship between differential GPX4 expression level in NGLY1-KO cells and resistance to ferroptotic cell death was also observed in other cell lines (HEK293T and MEF cells, Fig. S1).

3. Cell type specificity is not discussed or examined. Analysis of previous or public proteomic or transcriptomic data from systemic KO mice or NGLY1-deficient patient-derived cells could provide a new clue.

>Taking the reviewer's suggestion into consideration, we checked the “NGLY1 browser” (https://apps.embl.de/ngly1browser/) public database. GPX4 mRNA and protein expression levels in NGLY1 deficiency patients-derived fibroblasts and lymphoblastoid cells are reported. Interestingly, the expression levels of GPX4 varied between these cell types at both the mRNA and protein levels. In addition, we performed further experiments using HEK293T and MEF cells to further probe cell type specificity. As a result, NGLY1-KO and WT HEK293T cells showed similar levels of sensitivity to ferroptosis induction and GPX4 expression as HepG2 cells. However, Ngly1-KO and WT MEF cells showed the opposite result. The results from HEK293T and MEF cells are shown in Supplemental Figs. 1A-F.

4. Fig. 2G. The authors showed the increase in neutral lipids in KO cells. Please provide more detailed and reasonable explanation of the Fig. 2F and G data.

>We have added flow cytometry population mapping data (Figs. 2E-F), which shows an increased proportion of lipid peroxide-positive cells within the lipid-positive population in NGLY1-KO (KO-1) compared to WT HepG2 cells. This suggests that the elevated lipid peroxidation observed in NGLY1-KO HepG2 cells is not merely a consequence of increased lipid content. We have included this explanation in the main text (page 11, lines 260-264).

Minor points

5. Line 30-31. Structure of this sentence needs to be revised. “was” meant to be “were”?

>Taking the reviewer's comment carefully into consideration, we rechecked the sentences grammar. We are of the opinion that the subject the verb in question (“was”) refers to is “ferroptosis”. We remain of the believe that “was” is the grammatically correct verb and thus have chosen not to change the sentence in question (page 2, line 31).

6. Methods. Regarding centrifugation conditions, only “rpm” but not “g” is written in some places. Please define with g or rpm with the rotor name(s).

>Following the reviewer’s suggestion, we changed the description from “rpm” to “g”. (pages 8-9, lines 180, 188, 206)

7. L216. “sequence confirmation” needs to be revised with more detailed description. For instance, genomic sequencing.

>In accordance with the reviewer’s recommendation, we have changed the description from “sequence confirmation” to “genomic sequencing”. (page 9, line 222).

8. Fig. 2C. The aspect ratio of the graph seems strange and should be corrected.

>The aspect ratio of the graph in question has been corrected and the figure has been moved to Fig. 3D in the revised manuscript.

9. Typo

Line 102. Space is needed between value and unit, “1.0g/L”.

>The manuscript was revised as suggested (page 5, line 102).

Line 121. NP-40 should be spelled out when it first appears.

>The manuscript was revised as suggested (page 5, line 121).

Line 123. Space is needed for “pH8.0”.

>The manuscript was revised accordingly (page 5, line 123).

Line 257. Space is missing for 595nm and 520nm.

>The manuscript was revised as suggested (page 11, line 255).

Line 333. “are” should be “is”?

>The manuscript was revised as suggested (page 15, line 375).

Reviewer #2:

Comments:

The authors present an analysis of the consequences of loss of N-glycanase (encoded by the gene NGLY1) on cellular metabolism, with a particular focus on pathways implicated in oxidative stress and iron metabolism, using a human hepatocellular carcinoma cell line HepG2 cells. Loss of NGLY1 function is observed in the rare human genetic disorder NGLY1 Deficiency, characterized by global developmental delay, a hyperkinetic movement disorder, transient elevation of liver transaminases, peripheral neuropathy, and other features. The goal of this study is to probe the molecular mechanisms by which loss of N-glycanase drives these diverse features.

The manuscript is clear and presents some novel findings, notably reporting on disturbances in iron metabolism in NGLY1 deficient cells. However, my only real criticism is that the study represents a very modest increase in knowledge about N-glycanase’s role in regulating cellular metabolism and pathways involved in disease pathogenesis.

Summary – the manuscript presents a proteomic analysis of changes in protein levels in NGLY1 deficient HepG2 cells, noting a pronounced disturbance in pathways implicated in iron metabolism and ferroptosis, an iron-dependent oxidative form of cell death. The authors characterize changes in protein and mRNA levels associated with iron handling and ferroptosis, observe elevated iron influx, changes in key transferrin pathway members, increases in reactive oxygen species and lipid peroxidation.

Criticisms:

The manuscript details changes in markers/proteins/genes associated with ferroptosis regulation, but does not directly measure whether NGLY1 loss in HepG2 cells affects sensitivity to induction of ferroptosis (a form of cell death) as was reported earlier by Forcina et al (2022) https://doi.org/10.1073/pnas.2118646119. As result, statements such as “ferroptosis is prominently upregulated” (abstract), cells are in a “perpetual state of ferroptosis” (p. 4, 14) are misleading. More accurate would be to state that pathways implicated in ferroptosis regulation are altered. This is addressed somewhat in the discussion section, in which the authors note that the mutant cells are capable of cell growth and division, but may show “a ‘sublethal’ engagement of ferro

---

## [Decision Letter · Decision Letter 1]

20 Jun 2025

Dear Dr. Fujihira,

Thank you for submitting your manuscript to PLOS ONE. After careful consideration, we feel that it has merit but does not fully meet PLOS ONE’s publication criteria as it currently stands. Therefore, we invite you to submit a revised version of the manuscript that addresses the points raised during the review process.

**ACADEMIC EDITOR: **

2) Include the data presented in Supplemental figure 1 in the results section rather than the discussion section. 

We look forward to receiving your revised manuscript.

Kind regards,

Kostas Pantopoulos, PhD

Academic Editor

PLOS ONE

Journal Requirements:

Reviewers' comments:

Reviewer's Responses to Questions

**Comments to the Author**

Reviewer #1: (No Response)

Reviewer #2: All comments have been addressed

2. Is the manuscript technically sound, and do the data support the conclusions?

Reviewer #1: (No Response)

Reviewer #2: Yes

3. Has the statistical analysis been performed appropriately and rigorously?

Reviewer #1: (No Response)

Reviewer #2: Yes

4. Have the authors made all data underlying the findings in their manuscript fully available?

Reviewer #1: (No Response)

Reviewer #2: Yes

5. Is the manuscript presented in an intelligible fashion and written in standard English?

Reviewer #1: (No Response)

Reviewer #2: Yes

Reviewer #1: The authors sincerely addressed all my previous concerns. The revised paper is now ready for publication.

Reviewer #2: Response to revised submission:

The authors have conducted additional studies and have responded substantially to reviewers’ comments and criticisms.

Summary – the manuscript presents a proteomic analysis of changes in protein levels in NGLY1 deficient HepG2 cells, noting a pronounced disturbance in pathways implicated in iron metabolism and ferroptosis, an iron-dependent oxidative form of cell death. The authors characterize changes in protein and mRNA levels associated with iron handling and ferroptosis, observe elevated iron influx, changes in key transferrin pathway members, increases in reactive oxygen species and lipid peroxidation.

In my earlier review I noted that the manuscript details changes in markers/proteins/genes associated with ferroptosis regulation, but did not directly measure whether NGLY1 loss in HepG2 cells affects sensitivity to induction of ferroptosis. In response to this criticism, the authors examined the ferroptosis sensitivity of wt and NGLY1 deficient HepG2 cells and found that in this cellular context, NGLY1 deficient cells are in fact more resistant to ferroptosis induced by GPX4 inhibition compared to wild type cells. This results differs from an earlier study (Forcina et al (2022) https://doi.org/10.1073/pnas.2118646119) with a different cell type which found that NGLY1 deficient cells re more sensitive to ferroptosis induction. The authors also conducted additional studies and found that GPX4 protein and mRNA levels are increased in NGLY1 deficient HepG2 cells, which can account for their increased resistance to ferroptosis inducers, and likely represents a compensatory mechanism to allow cellular survival in the face of increased oxidative stress. The authors also assessed ferroptosis sensitivity in other cell types and examined publicly available datasets. They found that in some cellular contexts, NGLY1 loss is accompanied by increased GPX4 expression and resistance to ferroptosis induction, while in other contexts, NGLY1 loss does not lead to increased GPX4 levels, and NGLY1 deficient cells are more sensitive to ferroptosis induction. Overall, these data indicate that NGLY1 loss leads to disturbances in iron metabolism and oxidative stress, and that some cell types are able to develop compensatory mechanisms to tolerate oxidative stress without undergoing ferroptosis.

With these revisions, the manuscript provides a more through and detailed analysis of the consequences of NGLY1 loss on iron homeostasis, oxidative stress, and ferroptosis, and publication is supported.

Two revisions are suggested:

First, as noted in my earlier review, statements such as “ferroptosis is prominently upregulated” (abstract, line 31), and cells are in a “perpetual state of ferroptosis” (p. 4, lines 81-82; p. 15 line 379) are misleading. In fact, as shown here, NGLY1 deficient cells are MORE RESISTANT to ferroptosis induction by GPX4 inhibition. Thus, a much better statement would be to propose that NGLY1 cells are in a state of increased oxidative “pro-ferroptotic” stress and are able to tolerate this state of increased stress by compensatory upregulation of GPX4.

Second the data presented in Supplemental figure 1, showing that NGLY1 inactivation in different cell types has varying effects on ferroptosis sensitivity and correlates with varying changes in GPX4 expression, should be included in the results section rather than the discussion section.

-

**Do you want your identity to be public for this peer review?** For information about this choice, including consent withdrawal, please see our Privacy Policy

Reviewer #1: **Yes: ** Yasuhiko Kizuka

Reviewer #2: **Yes: ** Kevin Lee

---

## [Author Response · Author response to Decision Letter 2]

25 Jun 2025

Reviewer #2:

We wish to thank this reviewer for the valuable comments that are intended to improve our manuscripts.

Comments:

Response to revised submission:

The authors have conducted additional studies and have responded substantially to reviewers’ comments and criticisms.

Summary – the manuscript presents a proteomic analysis of changes in protein levels in NGLY1 deficient HepG2 cells, noting a pronounced disturbance in pathways implicated in iron metabolism and ferroptosis, an iron-dependent oxidative form of cell death. The authors characterize changes in protein and mRNA levels associated with iron handling and ferroptosis, observe elevated iron influx, changes in key transferrin pathway members, increases in reactive oxygen species and lipid peroxidation.

In my earlier review I noted that the manuscript details changes in markers/proteins/genes associated with ferroptosis regulation, but did not directly measure whether NGLY1 loss in HepG2 cells affects sensitivity to induction of ferroptosis. In response to this criticism, the authors examined the ferroptosis sensitivity of wt and NGLY1 deficient HepG2 cells and found that in this cellular context, NGLY1 deficient cells are in fact more resistant to ferroptosis induced by GPX4 inhibition compared to wild type cells. This results differs from an earlier study (Forcina et al (2022) https://doi.org/10.1073/pnas.2118646119) with a different cell type which found that NGLY1 deficient cells re more sensitive to ferroptosis induction. The authors also conducted additional studies and found that GPX4 protein and mRNA levels are increased in NGLY1 deficient HepG2 cells, which can account for their increased resistance to ferroptosis inducers, and likely represents a compensatory mechanism to allow cellular survival in the face of increased oxidative stress. The authors also assessed ferroptosis sensitivity in other cell types and examined publicly available datasets. They found that in some cellular contexts, NGLY1 loss is accompanied by increased GPX4 expression and resistance to ferroptosis induction, while in other contexts, NGLY1 loss does not lead to increased GPX4 levels, and NGLY1 deficient cells are more sensitive to ferroptosis induction. Overall, these data indicate that NGLY1 loss leads to disturbances in iron metabolism and oxidative stress, and that some cell types are able to develop compensatory mechanisms to tolerate oxidative stress without undergoing ferroptosis.

With these revisions, the manuscript provides a more through and detailed analysis of the consequences of NGLY1 loss on iron homeostasis, oxidative stress, and ferroptosis, and publication is supported.

Two revisions are suggested:

First, as noted in my earlier review, statements such as “ferroptosis is prominently upregulated” (abstract, line 31), and cells are in a “perpetual state of ferroptosis” (p. 4, lines 81-82; p. 15 line 379) are misleading. In fact, as shown here, NGLY1 deficient cells are MORE RESISTANT to ferroptosis induction by GPX4 inhibition. Thus, a much better statement would be to propose that NGLY1 cells are in a state of increased oxidative “pro-ferroptotic” stress and are able to tolerate this state of increased stress by compensatory upregulation of GPX4.

Second the data presented in Supplemental figure 1, showing that NGLY1 inactivation in different cell types has varying effects on ferroptosis sensitivity and correlates with varying changes in GPX4 expression, should be included in the results section rather than the discussion section.

In accordance with this reviewer’s suggestions, we have changed the description as below.

Revision to Figures

1. Previous Figs. S1A-F were moved to Figs. 4E-J.

2. Previous Figs. S2A-C were moved to Figs. S1A-C.

*In accordance with these revisions, associated Figure references in the text and associated Figure legends were revised (pages 29-30, lines 688, 691-696, 713-714).

Revision to Abstract

1. page 2, lines 31-32: The sentence was changed from “quantities in NGLY1-deficient HepG2 cells that suggested ferroptosis was prominently upregulated” to “quantities in NGLY1-deficient HepG2 cells, indicating that these cells are under “pro-ferroptotic” stress state”.

Revision to Introduction

1. page 4, lines 82-83: The sentence was changed from “NGLY1-KO HepG2 cells were found to be perpetually in a state of ferroptosis, a regulated form of cell death characterized by the iron-dependent accumulation of lipid peroxides” to “NGLY1-KO HepG2 cells were found to be in a state of pro-ferroptosis stress”.

2. page 4, lines 77-79: Due to the above change, the following sentence was added: Ferroptosis is a regulated form of cell death characterized by the iron-dependent accumulation of lipid peroxides (7, 11, 12).

Revision to Materials and Methods

Due to changes in the Figs, the contents previously described in the Supplemental methods section have been added to the Materials and Methods section.

1. Culture methods for HEK293T cells and MEF cells were added to “Cell Culture and Treatment” section (page 5, lines 104-106, 108).

1. Culture methods for HEK293T cells and MEF cells were added to “Cell Culture and Treatment” section (page 5, lines 104-106, 108).

2. The NGLY1-KO method in HEK293T cells was added to “Knockout of NGLY1 in HepG2 Cells” section (page 5, lines 109, 113, 117-118).

3. The ferroptosis inducer treatment method of HEK293T and MEF cells were added to “Ferroptosis inducer treatment of HepG2 cells” section (page 9, lines 211-214).

Revision to Results

1. The following sentences were added: Earlier reports indicated the heightened sensitivity to ferroptotic cell death of NGLY1-KO A549 cells as well as their diminished GPX4 protein levels (7). Therefore, we conducted the same experiments using cell lines other than HepG2 cells to examine the context-dependency of the impact of NGLY1 loss on ferroptotic susceptibility. Our results, obtained by using HEK293T and MEF cells, showed that NGLY1-KO HEK293T cells exhibit increased resistance to induced ferroptotic cell death, accompanied by elevated GPX4 protein levels when compared with WT HEK293T cells (Figs. 4 E, G, I). Conversely, Ngly1-KO MEF cells exhibited heightened sensitivity to induced ferroptotic cell death, characterized by reduced GPX4 protein levels in comparison to WT MEF cells (Figs. 4F, H, J). Taken together, these results highlight the context dependency of induced ferroptotic cell death resistance and GPX4 expression levels on NGLY1-KO/Ngly1-KO cell type. (pages 12-13, lines 306-315)

Revision to Discussion

1. page 15, lines 376-381: The text passage “Earlier reports indicated the heightened sensitivity to ferroptotic cell death of NGLY1-KO A549 cells as well as their diminished GPX4 protein levels (7). The public database “NGLY1 browser (https://apps.embl.de/ngly1browser/)” reports that NGLY1 deficiency patient-derived fibroblasts show increased GPX4 mRNA and reduced GPX4 protein levels, while patient-derived lymphoblastoid cells showed decreased GPX4 mRNA and protein levels. Moreover, preliminary results indicate that NGLY1-KO HEK293T cells exhibit increased resistance to induced ferroptotic cell death, accompanied by elevated GPX4 protein levels when compared with WT HEK293T cells. Conversely, Ngly1-KO MEF cells exhibited heightened sensitivity to induced ferroptotic cell death, characterized by reduced GPX4 protein levels in comparison to WT MEF cells (Figs. S1A-F). Taken together, these results highlight the context dependency of induced ferroptotic cell death resistance and GPX4 mRNA and protein expression levels on NGLY1-KO cell type.” was revised to “Regarding the context dependency of GPX4 expression levels on NGLY1-KO cell type, the public database “NGLY1 browser (https://apps.embl.de/ngly1browser/)” reports that NGLY1 deficiency patient-derived fibroblasts show increased GPX4 mRNA and reduced GPX4 protein levels, while patient-derived lymphoblastoid cells showed decreased GPX4 mRNA and protein levels. This observation also highlights the context dependency of GPX4 mRNA and protein expression levels on NGLY1-KO cell type.”.

2. page 16, lines 389-390: “perpetual state of ferroptosis” was changed to “increased oxidative “pro-ferroptotic” state and the compensatory upregulation of GPX4 to tolerate this state”.

---

## [Decision Letter · Decision Letter 2]

27 Jun 2025

Comparative Proteomics of HepG2 Cells Reveals NGLY1 as an Important Regulator of Ferroptosis Resistance and Iron Uptake

PONE-D-24-56966R2

Dear Dr. Fujihira,

We’re pleased to inform you that your manuscript has been judged scientifically suitable for publication and will be formally accepted for publication once it meets all outstanding technical requirements.

Kind regards,

Kostas Pantopoulos, PhD

Academic Editor

PLOS ONE

Additional Editor Comments (optional):

Reviewers' comments:

Reviewer's Responses to Questions

**Comments to the Author**

Reviewer #2: All comments have been addressed

2. Is the manuscript technically sound, and do the data support the conclusions?

Reviewer #2: Yes

3. Has the statistical analysis been performed appropriately and rigorously?

Reviewer #2: Yes

4. Have the authors made all data underlying the findings in their manuscript fully available?

Reviewer #2: Yes

5. Is the manuscript presented in an intelligible fashion and written in standard English?

Reviewer #2: Yes

Reviewer #2: (No Response)

**Do you want your identity to be public for this peer review?** For information about this choice, including consent withdrawal, please see our Privacy Policy

Reviewer #2: **Yes: ** Kevin Lee
